# GEOMETRY OF LIGHTNING SELF-ATTENTION: IDENTIFIABILITY AND DIMENSION

**Nathan W. Henry \***
University of Toronto
nathan.henry@mail.utoronto.ca

**Giovanni Luca Marchetti \***
Royal Institute of Technology (KTH)
glma@kth.se

**Kathlén Kohn \***
Royal Institute of Technology (KTH)
kathlen@kth.se

## ABSTRACT

We consider function spaces defined by self-attention networks without normalization, and theoretically analyze their geometry. Since these networks are polynomial, we rely on tools from algebraic geometry. In particular, we study the identifiability of deep attention by providing a description of the generic fibers of the parametrization for an arbitrary number of layers and, as a consequence, compute the dimension of the function space. Additionally, for a single-layer model, we characterize the singular and boundary points. Finally, we formulate a conjectural extension of our results to normalized self-attention networks, prove it for a single layer, and numerically verify it in the deep case.

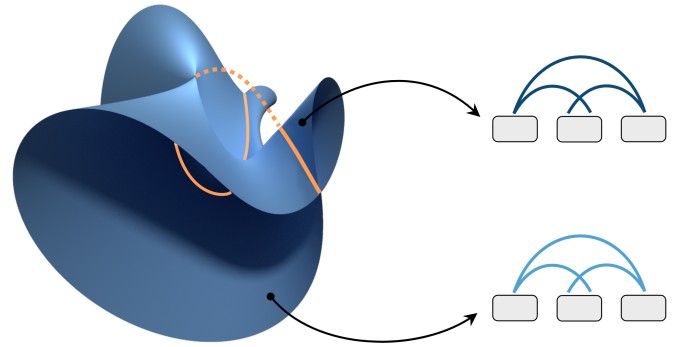

Figure 1: A slice of the space of lightning self-attention mechanisms.

## 1 INTRODUCTION AND RELATED WORK

The *self-attention mechanism* is the cornerstone of the Transformer – a modern machine learning architecture that is nowadays popular in a vast variety of domains, ranging from natural language processing (Vaswani et al., 2017), to vision (Dosovitskiy et al., 2020), to sound (Huang et al., 2018). In all of these domains, self-attention mechanisms have showcased outstanding performance due to their ability to model long-range dependencies within data sequences. *Lightning* self-attention mechanisms (Schlag et al., 2021) are standard variants where, differently from the original proposal, the attention weights are left un-normalized. As a result, the computational complexity of a forward pass is linear with respect to the sequence length, substantially improving on the quadratic complexity of the original model.

Despite their effectiveness, the theoretical understanding of self-attention mechanisms is superficial, and many aspects have yet to be clarified. In particular, understanding the geometry of function

---

*Equal contribution.

spaces defined by neural networks – typically referred to as *neuromanifolds* (Marchetti et al., 2025; Kohn, 2024; Calin, 2020) – is a fundamental challenge due to its intimate connection to several machine learning aspects, such as sample complexity and expressivity. Moreover, since neural networks learn by following a gradient flow over the neuromanifold, the geometry of the latter controls several aspects of the training dynamics (Trager et al., 2019). While neuromanifolds are well-understood for several architectures, such as fully-connected (Kileel et al., 2019; Kubjas et al., 2024) and convolutional networks (Kohn et al., 2022; 2023; Shahverdi et al., 2024), they have not been considered for self-attention mechanisms.

In this work, we study the geometry of neuromanifolds associated to lightning self-attention mechanisms. These models are of algebraic nature, since they are tri-linear in their weights and cubical in the input. This enables us to analyze neuromanifolds via ideas and tools from *algebraic geometry* – a rich field concerned with spaces defined by polynomial equations. In particular, it is possible to compute geometric quantities such as the *dimension* of the neuromanifold. The latter is a measure of expressivity of the underlying model. More concretely, it is intimately linked with sample complexity. According to the Fundamental Theorem of Learning, the dimension controls, linearly, the sample complexity of learnability (Shalev-Shwartz & Ben-David, 2014). This theory is typically formulated for (binary) classifiers[1], and the notion of dimension is a discrete one – the Vapnik–Chervonenkis (VC) dimension, specifically. In the continuous setting, the dimension of the neuromanifold is the natural analogue of the combinatorial VC dimension, and controls the sample complexity of learnability. An expression for sample complexity can be used both to select the appropriate model/architecture given an available dataset, and to collect appropriate amounts of data to train a given a model. This is especially important for attention-based models, that are nowadays popular in several domains, and are trained at extremely-large scales.

The dimension of the neuromanifold is related to the dual question of *identifiability* (Grigsby et al., 2023; Bona-Pellissier et al., 2023; Fefferman et al., 1994) – a problem concerned with characterizing the parameters corresponding to the same function. Geometrically, such parameters define *fibers* of the parametrization of the neuromanifold, and their (generic) dimension measures the difference between the dimension of the neuromanifold and the number of parameters. Therefore, characterizing fibers leads to an estimate of sample complexity which is more precise than the common practice of counting parameters. Moreover, understanding the fibers can be interesting beyond their relation to the dimension, since they control aspects of the training dynamics. Indeed, fibers induce invariances of the loss function which are data-independent, meaning that for any dataset, the loss will be constant for parameters within the same fiber. This gives rise to the phenomenon of flatness of the loss landscape (Zhao et al., 2022b), where minima are not isolated but instead belong to a continuous set. Even further, it is understood that the symmetries of the loss landscape control training dynamics as gradient directions must be orthogonal to fibers of the loss, which also induces a constraint on the Hessian (Kunin et al., 2020). This has recently been exploited to design optimizers that 'teleport' along fibers (Zhao et al., 2022a), improving learning efficiency.

## 1.1 SUMMARY OF CONTRIBUTIONS

Our core contribution is a description of the (generic) fibers of the parametrization of lightning self-attention networks. As a consequence, the expression for the dimension of the neuromanifold follows immediately. More specifically, our results are summarized as follows.

For a single layer of lightning self-attention (Section 3.1), we describe all the fibers, and additionally study various aspects of the geometry of the neuromanifold. Specifically, we prove that it is Euclidean closed and compute its singular and boundary points.

For a deep lightning self-attention network (Section 3.3), we compute the generic fibers. Our proof involves a reparametrization – which can be interpreted as introducing 'virtual weights' – and a subtle induction argument based on a closed-form algebraic expression for the network w.r.t. the new parameters. Assuming the network has a bottleneck architecture, we derive a formula for the dimension of the neuromanifold. In particular, the dimension is strictly lower than the number of parameters, with redundancies arising from a scaling symmetry, from inter-layer symmetries, and from the rank constraint on the attention weights.

---

[1]In this context, neuromanifolds are referred to as 'hypothesis spaces', and are usually considered in a combinatorial version.

Lastly, we study traditional self-attention by re-introducing the softmax normalization (Section 3.4). We prove that the parametrization of a single layer is generically one-to-one, and state a conjecture – verified via numerical experiments – for the generic fibers of deep self-attention networks.

## 2 LIGHTNING SELF-ATTENTION

Fix positive integers $d, d', a, t \in \mathbb{N}$ and matrices $Q, K \in \mathbb{R}^{a \times d}$, $V \in \mathbb{R}^{d' \times d}$. The latter are deemed *query weights*, *key weights*, and *value weights* respectively[2]. A self-attention mechanism is a map parametrized by $(Q, K, V)$ sending sequences of length $t$ of vectors in $\mathbb{R}^d$ to sequences of vectors in $\mathbb{R}^{d'}$. Here we consider the variant of self-attention mechanisms deemed *lightning*, which is computationally efficient and fully algebraic.

**Definition 1.** The *lightning self-attention mechanism* associated to the weights $W = (Q, K, V)$ is the map:

$$\begin{aligned}
\varphi_W \colon \mathbb{R}^{d \times t} &\rightarrow \mathbb{R}^{d' \times t} \\
(x_i)_{1 \leq i \leq t} &\rightarrow \left( \sum_{1 \leq j \leq t} x_j^\top \left( K^\top Q \right) x_i \, V x_j \right)_{1 \leq i \leq t}
\end{aligned} \tag{1}$$

Intuitively, every component $x_i$ of the input corresponds to a *token* and 'attends' bilinearly to every other $x_j$, producing a scalar weight $x_j^\top K^\top Q x_i \in \mathbb{R}$. These weights are used to aggregate the values $V x_j \in \mathbb{R}^{d'}$, obtaining the corresponding component of the output. The map defined by Equation 1 is tri-linear in $(Q, K, V)$ and homogeneous cubical in $x$. It is often convenient to write Equation 1 in matrix form: if $X = (x_i)_{1 \leq i \leq t}$ is interpreted as a $d \times t$ matrix, then:

$$\varphi_W(X) = V X X^\top K^\top Q X. \tag{2}$$

Moreover, we will often simplify the parametrization by introducing the *attention matrix*:

$$A = K^\top Q. \tag{3}$$

The latter will always be interpreted as a bilinear form $x^\top A y$.

Lightning attention mechanisms are variants of the traditional ones (Vaswani et al., 2017), where the attention weights $x_j^\top A x_i$ are normalized to a probability distribution across $j$ – see Section 3.4 for further details. The major practical advantage of the lightning variant is its computational efficiency with respect to the sequence length. Specifically, Equation 1 can be computed in $\mathcal{O}(t)$ time, while traditional self-attention mechanisms require $\mathcal{O}(t^2)$ time due to normalization. The improvement in efficiency motivates the term 'lightning'.

Self-attention mechanisms can be stacked in order to obtain a deep network architecture. To this end, fix positive integers $t, l, \mathbf{d} = (d_0, \ldots, d_l), \mathbf{a} = (a_1, \ldots, a_l)$, and weights $\mathbf{Q} = (Q_1, \ldots, Q_l), \mathbf{K} = (K_1, \ldots, K_l), \mathbf{V} = (V_1, \ldots, V_l)$, with $Q_i, K_i \in \mathbb{R}^{a_i \times d_i}$, $V_i \in \mathbb{R}^{d_i \times d_{i-1}}$.

**Definition 2.** A *deep self-attention network* associated to the weights $\mathbf{W} = (\mathbf{Q}, \mathbf{K}, \mathbf{V})$ is the map:

$$\varphi_{\mathbf{W}} \colon \mathbb{R}^{d_0 \times t} \rightarrow \mathbb{R}^{d_l \times t} \tag{4}$$

given by the composition $\varphi_{\mathbf{W}} = \varphi_{W_l} \circ \cdots \circ \varphi_{W_1}$.

Again, a deep self-attention network is homogeneous of degree $3^l$ in $x$. Based on this, we denote by $\mathrm{Sym}_{3^l}\left(\mathbb{R}^{d_0 \times t}, \mathbb{R}^{d_l \times t}\right)$ the vector space of homogeneous polynomial functions from $\mathbb{R}^{d_0 \times t}$ to $\mathbb{R}^{d_l \times t}$ of degree $3^l$ in all the output co-ordinates.

**Definition 3.** The *neuromanifold* of a deep self-attention network is the image of the parametrization map $\mathbf{W} \mapsto \varphi_{\mathbf{W}}$:

$$\mathcal{M}_{\mathbf{d}, \mathbf{a}} = \left\{ \varphi_{\mathbf{W}} \mid \mathbf{W} \in \mathbb{R}^{\sum_i d_i (2a_i + d_{i-1})} \right\} \subset \mathrm{Sym}_{3^l}\left(\mathbb{R}^{d_0 \times t}, \mathbb{R}^{d_l \times t}\right). \tag{5}$$

The neuromanifold is a semi-algebraic set by the Tarski-Seidenberg Theorem, meaning that it can be defined by a finite number of polynomial equalities and inequalities in $\mathrm{Sym}_{3^l}\left(\mathbb{R}^{d_0 \times t}, \mathbb{R}^{d_l \times t}\right)$.

---

[2]An alternative standard notation for $Q, K, V$ is $W_Q, W_K, W_V$. Our choice is motivated by better readability.

## 3 RESULTS

In this section, we study the neuromanifold of lightning attention networks, focusing on its parametrization and its dimension. Our core focus will be the description of the *fibers* of the parametrization map $\mathbf{W} \mapsto \varphi_{\mathbf{W}}$, meaning that we will describe the sets of weights that define the same function. More precisely, the fiber of $\varphi_{\mathbf{W}} \in \mathcal{M}_{\mathbf{d},\mathbf{a}}$ is the set

$$\{\mathbf{W}' \mid \varphi_{\mathbf{W}'} = \varphi_{\mathbf{W}}\}. \tag{6}$$

Once the fibers are understood, the dimension of the neuromanifold can be computed. To this end, it is actually sufficient to describe the *generic* fibers, i.e., the ones corresponding to 'almost all $\mathbf{W}$' or, more precisely, to $\mathbf{W}$ lying outside of the common zeros of a polynomial system. The co-dimension of such fibers is constant and coincides with the dimension of the neuromanifold.

In order to study the parametrization map and its fibers, it is convenient to split the problem by considering self-attention mechanisms as parametrized via the attention matrix. More precisely, we will think of self-attention mechanisms as parametrized, by abuse of notation, via weights $W = (A, V)$, where $A \in \mathbb{R}^{d \times d}$ is an arbitrary matrix, and will study the matrix multiplication map $(Q, K) \mapsto A = K^{\top} Q$ independently. We begin by considering the latter. When $a < d$, the matrix multiplication map is not surjective, since $A$ is constrained to have rank $\leq a$. In other words, the image of this map is the *determinantal variety* – defined as the set of matrices in $\mathbb{R}^{d \times d}$ of rank at most $a$. On the other hand, the fibers of the matrix multiplication map are subtle, since they are closely related to the problem of matrix factorization. Yet, it is still possible to describe the generic fibers. To this end, note that the map exhibits the following invariance: $K^{\top} Q = K'^{\top} Q'$, where $K' = CK$ and $Q' = C^{-\top} Q$ for an arbitrary invertible matrix $C \in \mathrm{GL}_a(\mathbb{R})$. Conversely, the following elementary result shows that this is the only symmetry of a generic fiber.

**Lemma 3.1.** *Suppose that $A = K^{\top} Q = K'^{\top} Q' = A'$ and that $\mathrm{rk}(A) = \mathrm{rk}(A') = a \leq d$. Then there exists a unique invertible matrix $C \in \mathrm{GL}_a(\mathbb{R})$ such that $K' = CK$ and $Q' = C^{-\top} Q$.*

*Proof.* See Appendix A.1 □

If follows from the above result that, for $a < d$, the generic fibers of the matrix multiplication map are isomorphic to $\mathrm{GL}_a(\mathbb{R})$, and therefore have dimension $a^2$. This recovers the well-known formula for the dimension of the determinantal variety, which coincides with $2ad - a^2 = a(2d - a)$.

### 3.1 SINGLE-LAYER IDENTIFIABILITY

We now describe completely the fibers of the parametrization of a lightning self-attention mechanism in terms of the attention matrix. By abuse of notation, we will write $\varphi_W$ for $W = (A, V)$. Firstly, note that it is always possible to rescale the weights without changing the function. That is, $(A, V)$ and $\left(\lambda A, \frac{1}{\lambda} V\right)$ belong to the same fiber for all $\lambda \in \mathbb{R} \setminus \{0\}$. Therefore, we will focus on the fibers up to rescaling.

**Theorem 3.2.** *Suppose $t \geq 2$. The fiber of $\varphi_W \in \mathcal{M}_{d,d',a}$ for a given $W = (A, V)$ is as follows:*

- *If $\mathrm{rk}(A) = \mathrm{rk}(V) = 1$, given tensor decompositions $A = k \otimes q$ and $V = h \otimes v$ for some $q, k, v \in \mathbb{R}^d \setminus \{0\}, h \in \mathbb{R}^{d'} \setminus \{0\}$, the fiber consists, up to rescaling, of $W$ and $W' = (v \otimes q, h \otimes k)$.*

- *If $\varphi_W = 0$, the fiber consists of $W' = (A', V')$ such that $A' = 0$ or $V' = 0$.*

- *Otherwise, the fiber consists only of rescalings of $W$.*

*Proof.* See Appendix A.2. □

Note that the second condition of Theorem 3.2 is negligible, i.e., it does not hold for almost all weights $W$, even under the constraint $\mathrm{rk}(A) \leq a$. Moreover, if $d, d' \geq 2$ or $d, a \geq 2$, the first condition is negligible as well. Therefore, the generic fibers are one-dimensional. As a consequence, it is possible to compute the dimension (in the sense of algebraic geometry) of the neuromanifold, even when parametrized via queries and keys, or equivalently, when $A$ is restricted to have rank $\leq a$.

**Corollary 3.3.** *Suppose that $t \geq 2$ and that $d, d' \geq 2$ or $d, a \geq 2$. The dimension of the neuromanifold is:*

$$\dim\left(\mathcal{M}_{d,d',a}\right) = \begin{cases} 2ad + dd' - a^2 - 1 & \text{if } a \leq d, \\ d^2 + dd' - 1 & \text{otherwise.} \end{cases} \tag{7}$$

*Proof.* The formula follows from the fact that the generic fibers of the parametrization are one-dimensional and that the dimension of the determinantal variety is $2\alpha d + dd' - \alpha^2$, where $\alpha = \min\{a, d\}$ (see discussion after Lemma 3.1). $\qquad\square$

### 3.2 Single-Layer Geometry

We will now describe the geometry of the neuromanifold of a single layer in more detail. We will return to the question of identifiability for deep networks in Section 3.3. Throughout this section, we assume that $t \geq 2$ and that either $d, d' \geq 2$ or $d, a \geq 2$.

**Theorem 3.4.** *The neuromanifold $\mathcal{M}_{d,d',a}$ is closed in the Euclidean topology. Its (relative) boundary points are those $\varphi_{(A,V)}$ of the form $A = k \otimes q$ and $V = h \otimes k$ for some $q, k \in \mathbb{R}^d, h \in \mathbb{R}^{d'}$. Moreover, $\mathcal{M}_{d,d',a}$ is not a smooth manifold: its singular points are the $\varphi_{(A,V)}$ satisfying $\mathrm{rk}(A)\mathrm{rk}(V) \leq 1$.*

*Proof.* This is an amalgamation of Corollaries A.1, A.4, and A.6 in Appendix A.3. $\qquad\square$

Figure 1 provides a visualization of $\mathcal{M}_{2,1,2}$ for $t = 2$. The latter has dimension 5 and is embedded in the 40-dimensional space $\mathrm{Sym}_3\left(\mathbb{R}^4, \mathbb{R}^2\right)$. The illustration shows a rendering of the neuromanifold in a 3-dimensional affine slice of the ambient space. This slice cuts a 2-dimensional section of the neuromanifold. The yellow line denotes the singular locus, whose dotted segment emerges by taking the closure in the Zariski topology.

The above result has several consequences, from a machine learning perspective and, in particular, in terms of learning dynamics. The fact that the neuromanifold is closed in the Euclidean topology implies that it contains its limit points. In particular, when the model is trained via a dynamical system – which is the case for gradient descent – any training trajectory that converges to an equilibrium in the ambient space will converge within the neuromanifold. Moreover, singularities of neuromanifolds are a central focus in Information Geometry, and specifically in Singular Learning Theory (Watanabe, 2009). According to the latter, singularities of neuromanifolds play a central role in deep learning, since the function learned by a neural network (via gradient descent) often corresponds to a singular point of the neuromanifolds [8]. In other words, singularities often attract the learning dynamics, resulting in a form of 'implicit bias' associated to the neural architecture. According to Theorem 3.4, singularities of the neuromanifold arise exactly when both $A$ and $V$ have rank $\leq 1$ (or vanish). Therefore, this result suggests an implicit bias in attention mechanism towards inferring (extremely) low-rank functions. Such bias has been empirically observed in a variety of neural architectures (Amari et al., 2006), and our result might suggest a mathematical explanation to this phenomenon, at least in the single-layer and lightning case.

### 3.3 Deep Networks

In this section, we completely describe the symmetries in the parameters of a generic function arising from a deep network of lightning self-attention layers. We show that there are only three symmetries: 1) each layer can be scaled by a constant, 2) the keys and queries within each layer can be scaled by an invertible matrix as in Lemma 3.1, and 3) the output of one layer can be scaled by an invertible matrix if the next layer cancels out this scaling. We now describe the latter type of parameter symmetry, before formally stating our main result in Theorem 3.7.

To this end, consider dimension vectors $\mathbf{d}, \mathbf{a}$ and weights $\mathbf{W} = (\mathbf{A}, \mathbf{V})$ of a network with $l$ layers. For $1 \leq i \leq l$, define:

$$L_i = \prod_{l-i \leq j < l} V_{l-j} \qquad\qquad M_i = L_{i-1}^\top A_i L_{i-1}. \tag{8}$$

Moreover, set $M_1 = A_1$. It follows immediately from Definition 2 that a deep self-attention network can be written in terms of $(\mathbf{M}, L)$, where $\mathbf{M} = (M_1, \ldots, M_l)$ and $L = L_l$. Therefore, we introduce

a further re-parametrization and write, with abuse of notation, $\varphi_{\mathbf{W}}$ for $\mathbf{W} = (\mathbf{M}, L)$. We call the parameters in this parametrization *virtual weights*.

This parametrization has symmetries. Namely, invertible matrices $C_i \in \mathrm{GL}_{d_i}(\mathbb{R})$ for $1 \leq i < l$, consider:

$$V_i' = C_i V_i C_{i-1}^{-1} \qquad\qquad A_{i+1}' = C_i^{-\top} A_{i+1} C_i^{-1}. \qquad (9)$$

In the above, we set $C_0$ and $C_l$ to the identity. Replacing $V_i$, $V_{i+1}$, $A_{i+1}$ with $V_i'$, $V_{i+1}'$, $A_{i+1}'$ does not alter the virtual weights $\mathbf{M}$ and $L$. Intuitively, this operation consists of transforming the output of the $i$-th layer and cancelling back the transformation in the next layer. The following result states that, for generic $\mathbf{V}$ and with an assumption on the dimensions $d_i$, the above procedure completely characterizes the generic fibers of the map $(\mathbf{A}, \mathbf{V}) \mapsto (\mathbf{M}, L)$.

**Lemma 3.5.** *Suppose that for some $\delta \in \mathbb{N}$ it holds that $d_i = \delta$ for all $0 < i < l$ and $d_0, d_l \geq \delta$. Let $(\mathbf{M}, L)$ be virtual weights (Equation 8) obtained from both $(\mathbf{A}, \mathbf{V})$ and $(\mathbf{A}', \mathbf{V}')$. If $\mathrm{rk}(L) = \delta$ then for every $i$ there exists a unique $C_i \in \mathrm{GL}_{d_i}(\mathbb{R})$ such that $V_i'$ and $A_{i+1}'$ are obtained from $V_i$ and $A_{i+1}$ via Equation 9.*

*Proof.* See Appendix A.4. $\qquad\qquad\qquad\qquad\qquad\qquad\qquad\qquad\qquad\qquad\qquad\qquad\qquad\square$

The assumption on the dimensions $d_i$ in the above result can be practically interpreted as a 'bottleneck' architecture. Specifically, this architecture is equivalent to including a low-dimensional embedding and a high-dimensional unembedding layer, which is common in the literature, especially in an interpretability context (Elhage et al., 2021).

Before computing the fibers of the re-parametrization, it is convenient to rephrase the latter. The following result provides a recursive expression of a deep attention network in terms of $(\mathbf{M}, L)$.

**Lemma 3.6.** *For $X \in \mathbb{R}^{d_0 \times t}$, define inductively:*

$$\begin{cases} D_0 = I \\ D_i = D_{i-1} X^\top M_i X D_{i-1} X^\top M_i^\top X D_{i-1}. \end{cases} \qquad (10)$$

*Then:*

$$\varphi_{\mathbf{W}}(X) = L_l X \left( \prod_{1 \leq i \leq l} D_{l-i} X^\top M_{l-i+1} X \right). \qquad (11)$$

*Proof.* See Appendix A.5. $\qquad\qquad\qquad\qquad\qquad\qquad\qquad\qquad\qquad\qquad\qquad\qquad\qquad\square$

The above result can be rephrased without recursion. Given $X = (x_i)_{1 \leq i \leq t}$, since $XX^\top = \sum_{1 \leq i \leq t} x_i x_i^\top$, Equation 11 states that $\varphi_{\mathbf{W}}(X)_k$ can be written for every $k$ as:

$$\sum_{k_1, \ldots, k_{\tilde{l}}} L_l x_{k_1} \left( \prod_{1 \leq j \leq \tilde{l}-1} x_{k_j}^\top \widetilde{M}_j x_{k_{j+1}} \right) x_{\tilde{l}}^\top A_1 x_k, \qquad (12)$$

where $\tilde{l} = (3^l - 1)/2$ and $\widetilde{M}_j$ is defined as follows. Let $\alpha_j$ be the index of the first non-zero digit from the right of $j$ in base 3 (i.e., its 3-adic valuation plus one). Then $\widetilde{M}_j$ equals to $M_{\alpha_j}$ if this digit is 1, and to $M_{\alpha_j}^\top$ if it is 2. We provide a diagram illustrating Equation 12 in Figure 2, where the matrices in the equation correspond, in order, to the regions between the lines.

We now discuss the fibers, as anticipated. Firstly, similarly to the one-layer case, $\mathbf{W} = (\mathbf{M}, L)$ can be rescaled without altering the corresponding function. More precisely, if $M_i' = \lambda_i M_i$ and $L' = \rho L$ for some $\lambda_i, \rho \in \mathbb{R} \setminus \{0\}$, then $\varphi_{\mathbf{W}'} = \varphi_{\mathbf{W}}$ if:

$$\prod_{1 \leq i \leq l} (\lambda_i)^{3^{l-i}} = \frac{1}{\rho}. \qquad (13)$$

Conversely, we now show that rescaling characterizes the generic fibers. The following is the main result of this work.

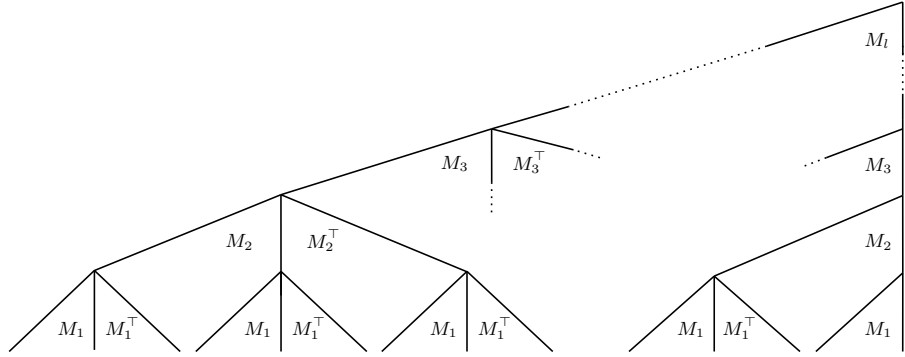

Figure 2: Diagrammatic illustration of Equation 12.

**Theorem 3.7.** *Let* $\mathbf{W} = (\mathbf{M}, L)$. *Suppose that* $t \geq 3$ *and:*

- *For* $1 \leq i \leq l$, $\mathrm{rk}(M_i) \geq 2$,

- *For* $1 < i \leq l$, $M_i$ *is not skew-symmetric*[3].

*Then the fiber of* $\varphi_{\mathbf{W}}$ *consists of rescalings of* $\mathbf{W}$.

*Proof.* The proof is highly technical. Here, we provide a short summary; for the full proof, see Appendix A.6. Using unique factorization of polynomials, we first show that, up to rescalings, the fiber consists of $\mathbf{W}' = (\mathbf{M}', L)$, where $M_i' = M_i + \Sigma_i$ and $\Sigma_i$ is a skew-symmetric matrix. We then proceed to show that $\Sigma_i = 0$ via an induction argument over $i$. The argument involves analyzing specific monomials where $\Sigma_i$ appears, and proving that most of them vanish due to the symmetries of the unrolled recursion tree in Figure 2. □

The conditions of the above theorem are generic if $d_i, a_i \geq 2$ for all $i$. To summarize, the theorem states that a generic function in the neuromanifold has precisely three types of symmetries in its parameters: 1) scaling each layer by a constant as in Equation 13, 2) scaling the keys and queries of each layer by invertible matrices as in Lemma 3.1, and 3) scaling the output of one layer by an invertible matrix and reverting this scaling in the next layer as in Equation 9. Similarly to the one-layer case, this results leads to the computation of the dimension of the neuromanifold.

**Corollary 3.8.** *Suppose that* $t \geq 3$, $a_i \geq 2$ *for all* $i$, *and for some* $\delta \geq 2$ *it holds that* $d_i = \delta$ *for all* $0 < i < l$ *and* $d_0, d_l \geq \delta$. *The dimension of the neuromanifold is:*

$$2\alpha_1 d_0 - \alpha_1^2 + \delta(d_0 + d_l) - \delta^2 - l + \sum_{1 < i \leq l} (2\alpha_i \delta - \alpha_i^2), \tag{14}$$

*where* $\alpha_i = \min\{a_i, d_{i-1}\}$.

*Proof.* See Appendix A.7. □

As discussed in Section 1, an exact expression for the dimension of the neuromanifold enables one to estimate the sample complexity of the model. While the latter is commonly measured as the number of parameters, the dimension – which constitutes the theoretically-correct estimate – can, sometimes, significantly differ from it. To illustrate this, in some instances the dimension of queries/keys is set to be equal to the embedding dimension (Vaswani et al., 2017), which translates to setting $\alpha_i = \delta$ for all $i$. Assuming $d_0 = d_l = \delta$, the number of parameters is $\frac{3}{2}d^2 l$, while according to Equation 14 the dimension is $d^2(l+1) - l$. Asymptotically, their ratio is $\frac{3}{2}$, i.e., in this case the number of parameters is 50% larger than the dimension.

---

[3]A square matrix $M$ is skew-symmetric if $M^\top = -M$.

### 3.4 TRADITIONAL SELF-ATTENTION

In this section, we briefly consider the case of traditional self-attention mechanisms, where the attention weights are normalized. We compute the fibers in the single-layer case and state a conjecture around the deep case.

In order to introduce the normalization, consider a map $\mathcal{S} \colon \mathbb{R} \to \mathbb{R}_{>0}$ that is injective and such that $\mathcal{S}(0) = 1$. A typical choice is $\mathcal{S}(x) = e^{x/\tau}$, where $\tau \in \mathbb{R}_{>0}$ is the temperature hyperparameter. The traditional self-attention mechanism is then defined for $W = (A, V)$ and $X \in \mathbb{R}^{d \times t}$ as:

$$\varphi_W(X) = \left( \frac{1}{\zeta_i} \sum_{1 \leq j \leq t} \mathcal{S}\left(x_i^\top A x_j\right) V x_j \right)_{1 \leq i \leq t} \qquad \zeta_i = \sum_{1 \leq k \leq t} \mathcal{S}\left(x_i^\top A x_k\right). \qquad (15)$$

Note that, for simplicity, we adhere to the convention of parametrizing via the attention matrix. Intuitively, in Equation 15 the attention weights $\mathcal{S}\left(x_i^\top A x_j\right)$ are normalized to sum to 1, forcing the model to 'distribute' its attention across the input components. The following result describes the fibers of the parametrization, analogously to Theorem 3.2.

**Theorem 3.9.** *Suppose that $t \geq 2$. If $\varphi_W = 0$, then $V = 0$. Otherwise, the fiber of $\varphi_W$ is a singleton $\{W\}$.*

*Proof.* See Appendix A.8. □

Therefore, differently from lightning self-attention, in this case, the parametrization is generically one-to-one.

We now consider the deep case. Note that even with normalization, deep attention networks can be reparametrized via $\mathbf{M}$ and $L$, as defined in Section 3.3. Therefore, the fibers of the parametrization will be unaffected by the symmetries inside the attention matrices from Lemma 3.1 and the transformations $C_i$ from Equation 9. However, this time no rescaling is possible. Therefore, we conjecture that the parametrization via $(\mathbf{M}, L)$ is generically one-to-one; in other words, that normalization only breaks the layer-wise scaling symmetry of the parametrization.

**Conjecture 3.10.** *For normalized deep self-attention networks, the generic fibers of the parametrization via $(\mathbf{M}, L)$ are singletons.*

In particular, suppose that for some $\delta$ it holds that $d_i = \delta$ for all $0 < i < l$ and $d_0, d_l \geq \delta$. Similarly to Corollary 3.8, the above conjecture implies that the dimension of the neuromanifold equals to:

$$2\alpha_1 d_0 - \alpha_1^2 + \delta(d_l + d_0) - \delta^2 + \sum_{1 < i \leq l} (2\alpha_i \delta - \alpha_i^2), \qquad (16)$$

which coincides with the dimension in the lightning case (Equation 14), summed with the number of layers $l$ due to the removal of scaling symmetries. This is an inconsequential difference for large models, where $l$ is significantly smaller than the number of parameters per layer.

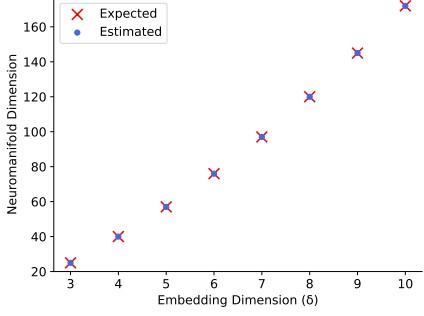

Figure 3: Plot of the estimated and expected dimensions of the neuromanifold as $\delta$ varies.

We provide empirical evidence for Conjecture 3.10. To this end, we implement a deep attention network with softmax normalization (i.e., $S(x) = e^x$), and estimate the dimension of its neuromanifold. The latter is a subtle problem since, differently from the lightning case, the neuromanifold is not a priori embedded in a finite-dimensional vector space. Therefore, we rely on a stochastic finite element approach by randomly sampling $N = 250$ input points in $\mathbb{R}^{d_0 \times t}$ from a normal distribution and restricting $\varphi_{\mathbf{W}}$ to this finite space. As a result, the neuromanifold is embedded in a $(N \times t \times d_l)$-dimensional vector space. Its dimension can then be computed as the rank of the Jacobian of the parametrization at a random parameter $\mathbf{W}$. This provides the correct result with probability 1 w.r.t. to the sampling of $\mathbf{W}$, with the only possible error coming from the discretization, which can be corrected for by increasing the number of input samples. Our `Python` code is available at a public repository[4]. The results are visualized in Figure 3 for a deep attention network with $l = 2$ layers, $t = 3$, $a_i = 2$ for all $i$, and $d_i = \delta$ varying from 3 to 10. The plot shows both the dimension estimated via the numerical approach ('Estimated') and the one computed via Equation 16 ('Expected'). The two values coincide for all $\delta$, confirming Conjecture 3.10 empirically.

## 4 Conclusions and Future Work

In this work, we have analyzed the geometry of neuromanifolds of lightning self-attention networks. In particular, we have provided a description of the fibers of the parametrization, and consequently computed the dimension of the neuromanifold for an arbitrary number of layers. Finally, we have formulated an analogous conjecture for traditional self-attention networks.

Our work represents a first step towards the mathematical understanding of neuromanifolds defined by attention networks. As such, it is subject to limitations and leaves several questions open. Specifically, the attention networks we consider are a simplified version of the ones deployed in practice, since we omit popular architectural variations (Brauwers & Frasincar, 2021). Two such variations, which are ubiquitous in contemporary Transformers, are skip connections and multiple heads. With both these additions, the lightning self-attention mechanism is still polynomial. Note that skip connections make the model non-homogeneous, which breaks the scaling symmetry in the parameterization. We believe that in this case the parameterization via $(\mathbf{M}, L)$ is generically one-to-one, similarly to the traditional case (Conjecture 3.10), which is also non-homogeneous. In contrast, including multiple attention heads introduces new symmetries due to permutation of heads, similarly to the permutation symmetries of traditional Multi-Layer Perceptrons (Kileel et al., 2019). Therefore, these two variations give rise to interesting phenomena in terms of symmetries of the parameterization, and define future directions that are worthy of exploration.

From a wider perspective, the research program of applying the tools offered by algebraic geometry to the field of deep learning remains open and worthy of exploration. Even further, going beyond the polynomial setting – e.g., addressing problems such as Conjecture 3.10 – is an even more general challenge that lies at the foundations of the theoretical understanding of deep learning.

## Acknowledgements

This work was partially supported by the Wallenberg AI, Autonomous Systems and Software Program (WASP) funded by the Knut and Alice Wallenberg Foundation.

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

## A    PROOFS OF THEORETICAL RESULTS

In this appendix, we include the proofs of the theoretical results in the main body of the paper.

### A.1    PROOF OF LEMMA 3.1

*Proof.* Since $a \geq \mathrm{rk}(Q) \geq \mathrm{rk}(A) = a$, we deduce $\mathrm{rk}(K) = a$, and similarly for $Q, K', Q'$. Therefore, without loss of generality, we can assume that $K = K_1 \oplus K_2$, where $K_1 \in \mathrm{GL}_a(\mathbb{R})$ and $K_2 \in \mathbb{R}^{a \times (d-a)}$, and similarly for $K', Q, Q'$. If follows immediately that exists a unique $C \in \mathrm{GL}_a(\mathbb{R})$ such that $K_1' = CK_1$. Since by hypothesis $K_1^\top Q_1 = K_1'^\top Q_1'$, we deduce $Q_1 = C^\top Q_1'$. But then $K_2^\top Q_1 = K_2'^\top Q_1' = K_2'^\top C^{-\top} Q_1$, implying that $K_2^\top = K_2'^\top C^{-\top}$ and, similarly, $Q_2 = C^\top Q_2'$, as desired. $\qquad\square$

### A.2    PROOF OF THEOREM 3.2

*Proof.* Suppose that $\varphi_W(X) = \varphi_{W'}(X)$ for all $X = (x_i)_{1 \leq i \leq t} \in \mathbb{R}^{d \times t}$, where $W' = (A', V')$. By comparing the terms containing only $x_1$ in the polynomial identity $\varphi_W(X)_1 = \varphi_{W'}(X)_1$, we obtain:

$$V x_1\, x_1^\top A x_1 = V' x_1\, x_1^\top A' x_1. \tag{17}$$

The above equation is a product of a quadratic form and a linear multivariate form. If $\mathrm{rk}(A) > 1$, the quadratic form is irreducible, and from the unique factorization property of polynomials it follows that, up to multiplicative scalars, $V$ coincides with $V'$, and the quadratic form associated to $A$ coincides with the one associated to $A'$. The same holds if $\mathrm{rk}(V) > 1$, since the sides of the above equation represent cubical polynomials that share a quadratic factor, but whose remaining linear factors are independent. In order to prove that $A$ and $A'$ coincide (up to scaling), consider the terms linear in $x_1$ and quadratic in $x_2$ in the identity $\varphi_W(X)_1 = \varphi_{W'}(X)_1$. This leads to:

$$V\!\!\!\!/\, x_2\, x_2^\top A x_1 = V'\!\!\!\!/\, x_2\, x_2^\top A' x_1. \tag{18}$$

It follows that the bilinear forms associated to $A$ and $A'$ coincide (up to scaling), as desired for the third case of the claim.

If instead $\mathrm{rk}(A) = \mathrm{rk}(V) = 1$, then it is possible to factor $A = k \otimes q$ and $V = h \otimes v$ for some vectors $q, k, v \in \mathbb{R}^d \setminus \{0\}$, $h \in \mathbb{R}^{d'} \setminus \{0\}$. Exchanging the role of $k$ and $v$ does not alter Equation 1, which provides the first case of the claim. Finally, since the ring of polynomials is an integral domain, if $\varphi_W(X) = 0$ then from Equation 17 it follows that either $A = 0$ or $V = 0$, as desired for the second case of the claim. $\qquad\square$

### A.3    PROOF OF THEOREM 3.4

To prove Theorem 3.4, we begin with the following key insight: a consequence of Theorem 3.2 is that it is possible to *projectify* the neuromanifold and its parametrization (with respect to the attention matrix). To this end, denote by $\mathbb{P}\mathcal{V} = \mathcal{V}/(\mathbb{R} \setminus \{0\})$ the projectification of a vector space $\mathcal{V}$. The second condition of Theorem 3.2 implies that the parametrization descends to a well-defined morphism:

$$\overline{\varphi} \colon \mathbb{P}\mathbb{R}^{d \times d} \times \mathbb{P}\mathbb{R}^{d \times d'} \to \mathbb{P}\mathrm{Sym}_3\left(\mathbb{R}^{d \times t}, \mathbb{R}^{d' \times t}\right). \tag{19}$$

We denote by $\mathbb{P}\mathcal{M}_{d,d',a}$ the image of the map $\overline{\varphi}$ – referred to as *projective neuromanifold*. The fact that $\overline{\varphi}$ is well-defined implies the following topological result.

**Corollary A.1.** *The neuromanifold $\mathcal{M}_{d,d',a}$ is closed in the Euclidean topology.*

*Proof.* A continuous map between a compact and a Hausdorff space is closed, and in particular has a closed image. These properties are satisfied by $\overline{\varphi}$, where both the domain and the codomain are equipped with the Euclidean topology over projective spaces. It follows that $\mathbb{P}\mathcal{M}_{d,d',a}$ is Euclidean closed in $\mathbb{P}\mathrm{Sym}_3\left(\mathbb{R}^{d \times t}, \mathbb{R}^{d' \times t}\right)$. Since the parametrization is homogeneous, the neuromanifold coincides with the affine cone of its projectification. Since affine cones of closed subspaces are closed, the claim follows. $\qquad\square$

Next, we characterize the lightning self-attention mechanisms that are singular in the neuromanifold, i.e., whose tangent space has a dimension greater than Equation 7. For that, we combine Theorem 3.2 with a computation of the critical points of the network parametrization map. Firstly, we remark that the last point in Theorem 3.2 implies that $\overline{\varphi}$ is generically one-to-one onto its image $\mathbb{P}\mathcal{M}_{d,d',a}$, i.e., it is *birational*.

**Corollary A.2.** *The map $\overline{\varphi}$ is birational onto its image $\mathbb{P}\mathcal{M}_{d,d',a}$, with special fibers of cardinality* 2.

*Proof.* This follows from the first and the last point of Theorem 3.2. □

Next, we compute the critical points of the parametrization. To this end, recall that a point is critical for a differentiable map if the differential at that point does not have maximal rank.

**Lemma A.3.** *A point $(A, V) = W$ is critical for the parametrization map $W \mapsto \varphi_W$ if and only if $A = k \otimes q$ and $V = h \otimes k$ for some $q, k \in \mathbb{R}^d$, $h \in \mathbb{R}^{d'}$.*

*Proof.* Given $W = (A, V)$, by computing the partial derivatives of $\varphi_\bullet$, we see that the differential sends a tangent vector $(\dot{A}, \dot{V})$ to $\dot{V}XX^\top AX + VXX^\top \dot{A}X$. The latter is interpreted as an element of the vector space $\text{Sym}_3\left(\mathbb{R}^{d \times t}, \mathbb{R}^{d' \times t}\right)$, which is identified with the tangent spaces of all of its elements. Since the dimension of the neuromanifold is one less than the dimension of the parameter space, the point $W$ is critical if and only if the kernel of the differential has dimension $> 1$. The kernel consists of those $(\dot{A}, \dot{V})$ that, for all $X$, satisfy

$$\dot{V}XX^\top AX = -VXX^\top \dot{A}X. \tag{20}$$

If $V = 0$, the differential vanishes whenever $\dot{V} = 0$; and similarly for $A$. This provides the first case of the claim. If $\text{rk}(A)\text{rk}(V) \geq 2$, Theorem 3.2 implies that the kernel consists of $(\dot{V} = -\lambda V, \dot{A} = \lambda A)$ for $\lambda \in \mathbb{R}$, and $W$ is therefore not critical. Finally, suppose that $\text{rk}(A) = \text{rk}(V) = 1$, and write $A = k \otimes q$ and $V = h \otimes v$ for some $q, k, v \in \mathbb{R}^d, h \in \mathbb{R}^{d'}$. In order to obtain a larger kernel, by Theorem 3.2, it is necessary that $v = k$ (up to scaling). In that case, for every $\dot{A}$ of the form $\dot{A} = p \otimes q$, we have that $\dot{V} = -h \otimes p$ is a solution to Equation 20, implying that $W = (A, V)$ is critical. □

By exploiting the fact that $\overline{\varphi}$ is birational with finite fibers, the above result leads us to the characterization of the singular points of $\mathcal{M}_{d,d',a}$.

**Corollary A.4.** *$\varphi_{(A,V)}$ is singular in the neuromanifold if and only if $\text{rk}(A)\text{rk}(V) \leq 1$.*

*Proof.* The morphism $\overline{\varphi}$ is finite and birational by Corollary A.2. For such a map, a standard fact from algebraic geometry (Kohn et al., 2017, Lemma 3.2) says that a point $\varphi_W$ is singular in the projective neuromanifold $\mathbb{P}\mathcal{M}_{d,d',a}$ if and only if either its fiber under $\overline{\varphi}$ has cardinality $\geq 2$ or $W$ is critical for $\overline{\varphi}$. By Lemma A.3 and Theorem 3.2, a point with $\text{rk}(A)\text{rk}(V) \leq 1$ is either critical or has a fiber of cardinality 2, and $\varphi_{(A,V)}$ is therefore singular in $\mathbb{P}\mathcal{M}_{d,d',a}$. Since $\mathcal{M}_{d,d',a}$ is the affine cone of $\mathbb{P}\mathcal{M}_{d,d',a}$, a point is singular in the projective neuromanifold if and only if the corresponding line is singular in the neuromanifold, from which the claim follows. □

Finally, we compute the boundary points of $\mathcal{M}_{d,d',a}$. We will see that they are precisely the critical values of the parametrization map. For that, we interpret the neuromanifold as a *Segre variety on linear forms*, as follows. Given a vector space $\mathcal{V}$, we consider its dual space $\mathcal{V}^*$ and the outer product of vectors of linear forms:

$$\tilde{\sigma} \colon (\mathcal{V}^*)^d \times (\mathcal{V}^*)^{d'} \to \text{Sym}_2(\mathcal{V})^{d \times d'},$$
$$(\alpha, \nu) \mapsto (\alpha_m \cdot \nu_n)_{1 \leq m \leq d, 1 \leq n \leq d'}.$$

We can projectify this map analogously to $\overline{\varphi}$, obtaining birational morphism (if $d > 1$ or $d' > 1$), but not an embedding. If $a < d$, we need to restrict the first factor of the map $\tilde{\sigma}$ to tuples of linear forms that span a subspace of rank at most $a$. We denote this space by $(\mathcal{V}^*)^d_{\leq a}$.

**Proposition A.5.** *The neuromanifold $\mathcal{M}_{d,d',a}$ is linearly isomorphic to the image of $\tilde{\sigma}$ restricted to $(\mathcal{V}^*)^d_{\leq a} \times (\mathcal{V}^*)^{d'}$, where $\mathcal{V} = \mathbb{R}^d$.*

*Proof.* We denote by $\alpha_m$ the linear form that takes the inner product with the $m$-th column of $A$ and by $\nu_n$ the inner product with the $n$-th row of $V$. We have seen in the proof of Theorem 3.2 that every $\varphi_W \in \mathcal{M}_{d,d',a}$ is uniquely determined by its term that is linear in $x_1$ and quadratic in $x_2$, which is

$$x_2^\top A x_1 V x_2 = \left( \sum_{m=1}^{d} x_{1,m}\, \alpha_m(x_2)\, \nu_n(x_2) \right)_{1 \le n \le d'}.$$

Since the $x_{1,m}$ are formal variables, this expression is the collection of all products of linear forms $\alpha_m \cdot \nu_n$ for $1 \le m \le d$ and $1 \le n \le d'$, which shows the claim. $\square$

In what follows, *relative boundary* refers to the set of points (in the Euclidean topology) in $\mathcal{M}_{d,d',a}$ that are limit points of sequences in $\overline{\mathcal{M}}_{d,d',a} \setminus \mathcal{M}_{d,d',a}$, where $\overline{\mathcal{M}}_{d,d',a}$ denotes the closure of the neuromanifold in the Zariski topology of its ambient space $\mathrm{Sym}_3\left( \mathbb{R}^{d \times t}, \mathbb{R}^{d' \times t} \right)$.

**Corollary A.6.** *$\varphi_W$ is on the relative boundary of the neuromanifold if and only if $W$ it is critical for the parametrization map.*

*Proof.* For simplicity, we write $\mathcal{M}$ for $\mathcal{M}_{d,d',a}$, and start by computing its Zariski closure $\overline{\mathcal{M}}$. The Zariski closure of a subset $S$ of $\mathbb{R}^N$ consists precisely of the real points in the complex Zariski closure of $S$ viewed as a subset of $\mathbb{C}^N$. We compute the complex Zariski closure of $\mathcal{M}$ by considering the parametrization map $\overline{\varphi}$ in (19) over $\mathbb{C}$ instead of over $\mathbb{R}$. That map $\overline{\varphi}_\mathbb{C}$ is also a well-defined morphism between projective spaces. The main theorem on projective varieties (Hartshorne, 2013, II, §4, Theorem 4.9) states that the image $\mathbb{P}\mathcal{M}^\mathbb{C}$ of $\overline{\varphi}_\mathbb{C}$ is Zariski closed. Hence, the real Zariski closure $\overline{\mathcal{M}}$ consists precisely of the real points $\varphi_{(A,V)}$ with $A \in \mathbb{C}^{d \times d}, V \in \mathbb{C}^{d \times d'}$.

Next, we compute the points in the complement $\overline{\mathcal{M}} \setminus \mathcal{M}$. Using the identification in Proposition A.5, points in that complement look like real matrices with entries $\alpha_m \cdot \nu_n$, where $\alpha_m, \nu_n$ are complex linear forms, and no real parametrization exists. So one of the $\alpha_m$ is non-real. Since each of the $\alpha_m \cdot \nu_n$ is real, every $\nu_n$ has to be the complex conjugate $\overline{\alpha_m}$ times a real scalar. Analogously, all $\alpha_r$ have to be $\overline{\nu_n}$ times a real scalar, i.e., they need to coincide with $\alpha_m$ up to scaling. This shows that $\overline{\mathcal{M}} \setminus \mathcal{M}$ consists of $\varphi_{(A,V)}$ with $A = k \otimes q$ and $V = h \otimes \overline{k}$ for some $q \in \mathbb{R}^d \setminus \{0\}, h \in \mathbb{R}^{d'} \setminus \{0\}, k \in \mathbb{C}^d \setminus \mathbb{R}^d$. In the limit, sequences of $W = (A,V)$ in this form can become real if either one among $A$ and $V$ vanish, or the complex conjugated pair $k, \overline{k}$ becomes the same real vector $k = \overline{k}$. $\square$

## A.4 Proof of Lemma 3.5

*Proof.* Note that $V_i$ is invertible for every $1 < i < l$ by the rank hypothesis. Moreover, since $L = \prod_{0 \le j < l} V_{l-j} = \prod_{0 \le j < l} V'_{l-j}$, it is possible to apply Lemma 3.1 by splitting the latter factorization of $L$ at the first and last layer. This way, we obtain matrices $C_i \in \mathrm{GL}_\delta(\mathbb{R})$ for $1 \le i \le l$ such that $V'_i = C_i V_i C_{i-1}^{-1}$. But then we have:

$$L'^\top_{i-1} A'_i L'_{i-1} = L^\top_{i-1} C^\top_{i-1} A'_i C_{i-1} L_{i-1} = L^\top_{i-1} A_i L_{i-1}. \tag{21}$$

Since $L_i$ is surjective by the rank hypothesis, we conclude that $C^\top_{i-1} A'_i C_{i-1} = A_i$. $\square$

## A.5 Proof of Lemma 3.6

*Proof.* For $0 \le i \le l$, denote by $X_i$ the output of the $i$-th layer of the network. In other words, $X_0 = X$ and $X_i = \varphi_{W_i}(X_{i-1}) = V_i X_{i-1} X_{i-1}^\top A_i X_{i-1}$ for $i > 0$. We wish to prove that for all $i$:

$$X_i = L_i X \prod_{1 \le j \le i} \left( D_{i-j} X^\top M_{i-j+1} X \right). \tag{22}$$

Denote by $Y_i$ the right-hand side of Equation 22. We have:

$$Y_i = \underbrace{L_i}_{V_i L_{i-1}} X D_{i-1} X^\top \underbrace{M_i}_{L_{i-1}^\top A_i L_{i-1}} X \prod_{1 \leq j < i} D_{i-1-j} X^\top M_{i-j} X \qquad (23)$$

$$= V_i \, L_{i-1} X D_{i-1} \left(L_{i-1} X\right)^\top A_i \underbrace{L_{i-1} X \prod_{1 \leq j < i} D_{i-1-j} X^\top M_{i-j} X}_{Y_{i-1}}. \qquad (24)$$

Therefore, in order to show that $Y_i$ satisfies the same recurrence relation as $X_i$, we need to prove that $L_i X D_i \left(L_i X\right)^\top = Y_i Y_i^\top$ for all $i$, which in turn reduces to show that $D_i = Z_i Z_i^\top$, where:

$$Z_i = \prod_{1 \leq j \leq i} \left(D_{i-j} X^\top M_{i-j+1} X\right). \qquad (25)$$

To this end, since $Z_i = D_{i-1} X^\top M_i X Z_{i-1}$, we have:

$$Z_i Z_i^\top = D_{i-1} X^\top M_i X Z_{i-1} Z_{i-1}^\top X^\top M_i^\top X D_{i-1}^\top. \qquad (26)$$

If we assume inductively that $Z_{i-1} Z_{i-1}^\top = D_{i-1}$, and using the fact that $D_i$ is a symmetric matrix (i.e., $D_{i-1}^\top = D_{i-1}$), the above recurrence relation coincides with the one defining $D_i$ in Equation 11. This implies that $Z_i Z_i^\top = D_i$, as desired. $\qquad \square$

## A.6 Proof of Theorem 3.7

*Proof.* Pick weights $\mathbf{W}, \mathbf{W}'$ such that $\varphi_{\mathbf{W}}(X) = \varphi_{\mathbf{W}'}(X)$ for all $X = (x_i)_{1 \leq i \leq t} \in \mathbb{R}^{d_0 \times t}$. Our proof strategy will involve considering the polynomial identity

$$\varphi_{\mathbf{W}}(X)_1 = \varphi_{\mathbf{W}'}(X)_1 \qquad (27)$$

and comparing monomial terms arising from Equation 12 with specific degrees in the $x_i$'s.

To this end, Equation 12 implies that all such terms contain $x_1$. Even further, the unique term in $\varphi_{\mathbf{W}}(X)_1$ that is linear in $x_1$ and of degree $3^l - 1$ in $x_2$ can be written as:

$$L_l x_2 \left(\prod_{2 \leq i \leq l} \left(x_2^\top M_i x_2\right)^{3^{l-i}}\right) \left(x_2^\top A_1 x_2\right)^{3^{l-1} - 1} x_2^\top A_1 x_1. \qquad (28)$$

Put simply, the above expression is a product of a linear form in $x_2$, several quadratic forms in $x_2$, and a bilinear form in $x_1$ and $x_2$. Moreover, the quadratic forms are coprime by hypothesis since $\mathrm{rk}(M_i) > 1$ for $i > 1$. By comparing this term with the corresponding term on the right-hand side of Equation 27, from the unique factorization property of polynomials, it follows that up to a multiplicative scalar, the following hold:

- $L_l$ coincides with $L_l'$,

- $A_1$ coincides with $A_1'$,

- The quadratic forms associated to $M_i$ and $M_i'$ coincide for all $2 \leq i \leq l$.

The last condition above implies that for every $2 \leq i \leq l$ there exists a skew-symmetric matrix $\Sigma_i$ such that $M_i'$ coincides with $M_i + \Sigma_i$ up to a multiplicative scalar. Since Equation 12 is multi-linear in each occurrence of $M_j$, by substituting $M_j' = M_j + \Sigma_j$ in $\varphi_{\mathbf{W}'}(X)_1$ for all $j$, we obtain a sum of expressions in the form of Equation 12, but where an arbitrary number of occurrences of $M_j$ has been replaced by $\Sigma_j$. The only expression with no replacement coincides with $\varphi_{\mathbf{W}}(X)_1$. Therefore, Equation 27 reduces to a vanishing sum of copies of Equation 12, where at least one $M_j$ has been replaced with $\Sigma_j$.

In order to conclude, we wish to show that $\Sigma_i = 0$ for $i \geq 2$. We will proceed by induction on $i$. Specifically, given $i$ and assuming that $\Sigma_j = 0$ for $j < i$, we will show that $\Sigma_i = 0$. To this end, consider the monomial terms in $\varphi_{\mathbf{W}'}(X)_1$ of degree 1 in $x_1$, of degree $3^{i-1} - 1$ in $x_3$, and

of remaining degree in $x_2$. This is possible since we assume $t \geq 3$. From the discussion above, it follows that Equation 27 reduces to a vanishing sum of copies of Equation 12, where an arbitrary (non-zero) number of occurrences of $\widetilde{M}_j = M_{\alpha_j}$, with $\alpha_j \geq i$, has been replaced with $\Sigma_{\alpha_j}$ (and similarly with transposes), and where $k_j \in \{2, 3\}$ for all $1 \leq j \leq \tilde{l} = (3^l - 1)/2$.

We will now argue that several terms cancel due to the symmetries of Equation 12, illustrated in figure 4. Namely, consider a monomial term with some multi-index $k_\bullet$ containing a factor of the form $x_{k_j}^\top \Sigma_{\alpha_j} x_{k_{j+1}}$ for some $j$, with $\alpha_j > i$. Since $x_3$ cannot appear more than $3^{i-1} - 1$ times in the monomial, there exists a $\underline{j} \leq (3^{i-1} - 1)/2$ such that $k_{j-\underline{j}} = k_{j+\underline{j}+1} = 2$. Moreover, due to the inductive hypothesis, $\widetilde{M}_{j'}$ coincides with $M_{\alpha_{j'}}$ or with its transpose for all $j - \underline{j} \leq j' \leq j + \underline{j}$ since $\alpha_{j'} < i$. Consider the multi-index $k'_\bullet$ such that $k'_{j-j'} = k_{j+j'+1}$ for all $-\underline{j} \leq j' \leq \underline{j}$, and $k'_{j'} = k_{j'}$ for all the other $j'$. Intuitively, $k'_\bullet$ 'reflects' $k_\bullet$ locally around $j$ – see Figure 4 for an illustration. By construction, the monomial corresponding to $k'_\bullet$ is identical to the one corresponding to $k_\bullet$, except for the term

$$x_{k'_j}^\top \Sigma_{\alpha_j} x_{k'_{j+1}} = x_{k_{j+1}}^\top \Sigma_{\alpha_j} x_{k_j} = -x_{k_j}^\top \Sigma_{\alpha_j} x_{k_{j+1}}. \tag{29}$$

Therefore, these two monomials cancel each other out.

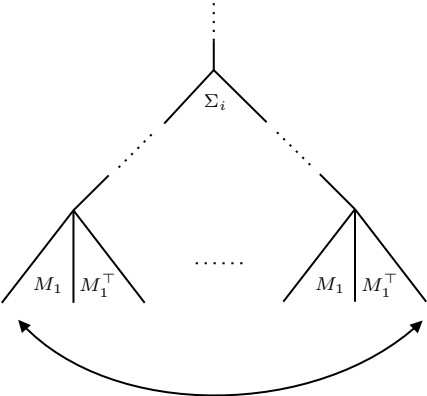

Figure 4: Diagrammatic illustration of the symmetry involved in the cancellation argument.

After the above cancellation argument, we are left with monomials containing occurrences of $\Sigma_{\alpha_j}$ only for $\alpha_j = i$. In fact, the cancellation argument still applies verbatim to these cases, except for when $j = (3^l - 3^{i-1})/2$ and $k_{j-j'} \neq k_{j+j'+1}$ for all $j' \leq (3^{i-1} - 1)/2$. In this case, due to the presence of $x_1$, the monomials corresponding to $k_\bullet$ and $k'_\bullet$ are not opposite, but contain different factors given by $x_3^\top A_1 x_2 x_2^\top A_1 x_1$ and $-x_3^\top A_1 x_1 \, x_2^\top A_1 x_2$, respectively. At the end of the day, Equation 27 reduces to an expression of the form:

$$x_3^\top \Sigma_i x_2 \; \left( x_3^\top A_1 x_2 x_2^\top A_1 x_1 - x_3^\top A_1 x_1 \, x_2^\top A_1 x_2 \right) \sum_{h_\bullet} p((x_{h_j})_j) = 0, \tag{30}$$

where $h_\bullet$ is an opportune multi-index with values in $\{2, 3\}$, and $p$ is a polynomial consisting of products of bilinear/quadratic forms associated to the $M_j$'s, and a multi-variate linear factor associated to $L_l$. The right-most factor in Equation 30 is not the zero polynomial (in the variables $x_2$ and $x_3$) since, by imposing the condition $x_2 = x_3$, it becomes (up to a multiplicative scalar) a product of quadratic forms associated to the $M_j$'s. The latter are non-vanishing since $M_j$ is not skew-symmetric for all $j$ by hypothesis.

We wish to prove that the second factor in Equation 30 is non-vanishing as well. To this end, note that the coefficients of that factor, seen as a polynomial in $x_1, x_2, x_3$, are of the form

$$(A_1)_{\alpha,\beta}(A_1)_{\gamma,\delta} + (A_1)_{\alpha,\gamma}(A_1)_{\beta,\delta} - (A_1)_{\alpha,\delta}(A_1)_{\beta,\gamma} - (A_1)_{\alpha,\delta}(A_1)_{\gamma,\beta} \tag{31}$$

for $1 \leq \alpha, \beta, \gamma, \delta \leq d_0$. Suppose by contradiction that the above expression vanishes for all $\alpha, \beta, \gamma, \delta$. When $\beta = \gamma$, Equation 31 coincides with (twice) the determinant of an arbitrary $2 \times 2$

minor of $A_1$ intersecting the diagonal. If $(A_1)_{\beta,\beta} \neq 0$ for some $\beta$, then from the Kronecker's bordered matrix theorem it follows that $A_1$ has rank 1, contradicting the hypothesis $\mathrm{rk}(A_1) \geq 2$. We conclude that the diagonal entries of $A_1$ are zero, and, when $\beta = \gamma$, Equation 31 reduces to $(A_1)_{\alpha,\beta}(A_1)_{\beta,\delta} = 0$. Therefore, if $(A_1)_{\alpha,\beta} \neq 0$ for some $\alpha \neq \beta$, then the $\beta$-th row of $A_1$ must vanish. But then Equation 31 reduces (up to sign) to the determinant of an arbitrary $2 \times 2$ minor intersecting $(\alpha, \beta)$, and we again obtain a contradiction from Kronecker's bordered matrix theorem.

In conclusion, the left-most factor in Equation 30 must vanish, meaning that $\Sigma_i = 0$, as desired. $\quad\square$

### A.7 PROOF OF COROLLARY 3.8

*Proof.* Recall that the dimension of the determinantal variety of $d_{i-1} \times d_{i-1}$ matrices of rank at most $a_i$ is $2\alpha_i d_{i-1} - \alpha_i^2$, and therefore the space of parameters $(\mathbf{A}, \mathbf{V})$ has dimension

$$2\alpha_1 d_0 - \alpha_1^2 + \delta(d_0 + d_l) + (l-2)\delta^2 + \sum_{1 < i \leq l} (2\alpha_i \delta - \alpha_i^2). \tag{32}$$

Moreover, by Lemma 3.5 the generic fibers of the re-parametrization $(\mathbf{A}, \mathbf{V}) \mapsto (\mathbf{M}, L)$ have dimension $(l-1)\delta^2$, while by Theorem 3.7 the generic fibers with respect to $(\mathbf{M}, L)$ have dimension $l$ (recall the constraint on rescaling given by Equation 13). Since the dimension of the image of a map coincides with the co-dimension of the generic fibers, the result follows. $\quad\square$

### A.8 PROOF OF THEOREM 3.9

*Proof.* Suppose that $\varphi_W(X) = \varphi_{W'}(X)$ for all $X = (x_i)_{1 \leq i \leq t} \in \mathbb{R}^{d \times t}$, where $W' = (A', V')$. We assume that $V \neq 0$, since the case $\varphi_W = 0$ follows immediately. Due to normalization, if $x_i = x_j$ for all $i, j$, then $\varphi_W(X)_1 = V x_1 = V' x_1 = \varphi_{W'}(X)_1$, implying $V = V'$. If instead $x_i = 0$ for $i > 1$, since $\mathcal{S}(0) = 1$, we obtain:

$$\frac{\mathcal{S}\left(x_1^\top A x_1\right)}{\mathcal{S}\left(x_1^\top A x_1\right) + t - 1} \cancel{V x_1} = \frac{\mathcal{S}\left(x_1^\top A' x_1\right)}{\mathcal{S}\left(x_1^\top A' x_1\right) + t - 1} \cancel{V' x_1} \tag{33}$$

Since $\mathcal{S}$ is injective, we deduce that the quadratic form associated to $A$ coincides with the one associated to $A'$. In order to prove that $A = A'$, suppose that $x_i = x_j$ for $i, j \geq 2$, obtaining:

$$\frac{\mathcal{S}\left(x_1^\top A x_1\right) V x_1 + \mathcal{S}\left(x_1^\top A x_2\right) V x_2}{\mathcal{S}\left(x_1^\top A x_1\right) + (t-1)\mathcal{S}\left(x_1^\top A x_2\right)} = \frac{\mathcal{S}\left(x_1^\top A' x_1\right) V' x_1 + \mathcal{S}\left(x_1^\top A' x_2\right) V' x_2}{\mathcal{S}\left(x_1^\top A' x_1\right) + (t-1)\mathcal{S}\left(x_1^\top A' x_2\right)}. \tag{34}$$

After elementary algebraic manipulations using the fact that $x_1^\top A x_1 = x_1^\top A' x_1$ and $V = V'$, the above equation reduces to:

$$(t-1)\left(\mathcal{S}\left(x_1^\top A' x_2\right) - \mathcal{S}\left(x_1^\top A x_2\right)\right) V x_1 = \left(\mathcal{S}\left(x_1^\top A' x_2\right) - \mathcal{S}\left(x_1^\top A x_2\right)\right) V x_2. \tag{35}$$

Since $V \neq 0$, $V x_1 \neq V x_2$ for generic $x_1, x_2$. Therefore, $\mathcal{S}\left(x_1^\top A' x_2\right) - \mathcal{S}\left(x_1^\top A x_2\right)$ must vanish generically, implying $A = A'$ due to the injectivity of $\mathcal{S}$, as desired. $\quad\square$

