# OpenReview forum: "Geometry of Lightning Self-Attention: Identifiability and Dimension"
_ICLR.cc/2025/Conference — ICLR 2025 Poster_

### Official Review · Reviewer_SY47 · 2024-11-02

**Soundness:** 3
**Presentation:** 2
**Contribution:** 2
**Rating:** 6
**Confidence:** 2

**Summary:**

This paper studies the neuromanifold, the image of the parametrization map that maps network weights to the corresponding input-output map of the network, of single-layer and multi-layer linear-self-attention networks. Specifically, the dimensions of these neuromanifolds are computed, and the fibers of the parametrization of these neuromanifolds are discussed.

**Strengths:**

Analysis of linear self-attention networks with tools from algebraic geometry

**Weaknesses:**

The reviewer is incapable of evaluating the technical contributions of this paper. However, the reviewer has concerns about the significance of the results to ML community and the paper's lack of accessibility to a broader ML audience.

1. This paper computes the intrinsic dimensions of the neuromanifolds for single- and multi-layer self-attention networks, and they are of the same scale as the number of parameters. (single-layer: $2ad+dd'$ v.s. $2ad+dd'-a^2-1$, and multi-layer 8.2b v.s. 8.6b, in an LLM example). The reviewer understands the research effort behind computing the intrinsic dimensions and this review does not mean to downplay its importance; but still, the results do not provide a better characterization of the complexity of the function space than simple parameter counting.

2. This paper also studies the fiber of the parametrization, the boundary of the neuromanifold, critical points of the parametrization map, etc. They are presumably important characteristics to be studied in algebraic geometry, but are they important to the deep learning community? Most of the results are stated without discussing their relevance to the training or generalization of these linear self-attention networks, except for Corollary 3.5: "Therefore, any training trajectory that converges to an equilibrium in the ambient space will converge within the neuromanifold", which is, in reviewer's opinion, too vague to imply anything useful.

Therefore, while the results presented in this paper may be of interest to the algebraic geometry community, the reviewer does not see it getting well appreciated by the ML community in its current form.

**Questions:**

See "Weaknesses". The authors do not need to spend too much time responding to this review, rather, they should revise the paper to be more accessible to a broader audience and motivate the results from a more practical perspective.

---

> ### Author Response · Authors · 2024-11-18
>
> We thank the reviewer for the constructive feedback.
>
> We acknowledge that our paper, in the submitted form, is lacking motivations and discussion, especially from a practical perspective. To this end, we have re-uploaded a new version of the manuscript, where we have elaborated on several points including the importance of dimension and its relation to sample complexity, the role of fibers and singularities in the learning dynamics, a more accessible explanation of our main result (Theorem 3.17), and specific ideas for future work. All these points are also summarized in the collective comment we posted above.
>
> We additionally wish to comment on the raised weakness raised about the relation between number of parameters and dimension of the neuromanifold. We argue that the difference between the two can potentially be quite large, especially for attention mechanisms, since, differently from other models, their parametrization exhibits plenty of symmetries – i.e., symmetries between queries and keys of the same layer, and between attention weights and values of successive layers. This applies to both the lightning case and, assuming Conjecture 3.19, to the traditional one. Therefore, the dimension of the neuromanifold can differ significantly from the number of parameters. To illustrate this, there are instances where the dimension of queries/keys equal to the embedding dimension (see the original paper [1]), i.e., $\alpha_i = \delta$ in Corollary 3.8. Assuming $d_0 = d_l = \delta $ as well, the number of parameters is $\frac{3}{2}d^2 l $, while according to our formula the dimension is $d^2( l + 1) - l$. Asymptotically, their ratio is $\frac{3}{2}$, i.e., the number of parameters is 50 \% more than the dimension.  We have added this discussion in the re-uploaded version of the manuscript in lines 369-377.
>
>  We hope that these discussions help to clarify the motivation and significance of our results.
>
> [1] Vaswani et al., Attention is all you need, 2017.

---

> > ### Comment · Reviewer_SY47 · 2024-11-21
> >
> > Thanks for the revision. I changed my rating accordingly.

---

### Official Review · Reviewer_qe2M · 2024-11-03

**Soundness:** 4
**Presentation:** 2
**Contribution:** 2
**Rating:** 6
**Confidence:** 2

**Summary:**

This paper studies the geometry of self-attention networks, focusing on "lightning" self-attention mechanisms. The paper offers a theoretical analysis of lightening attention using algebraic geometry, focusing on dimension of the neurmoanifold and the identifiability of deep attention. Key contributions include a description of the fibers of the parametrization map and the computation of neuromanifold dimensions. Additionally, the authors extend their findings to traditional, normalized self-attention networks (essentially with softmax activation) and present a conjecture, supported by empirical evidence, regarding their fibers of parametrization.

**Strengths:**

The paper presents a novel and rigorous theoretical analysis of undoubtedly important topics, i.e., attention mechanisms. The authors use algebraic geometry to develop a solid theoretical understanding of the neuromanifold of neural networks using attention mechanisms. Although the analysis is offered in a simplified context, that is standard and inevitable for a rigorous theoretical analysis and does not reduce the importance of their contribution. I believe the introduction motivates the study quite well (though it could be improved). Understanding the dimension and the topology of the neuromanifold is a consequential result, as it relates to the learning dynamics of transformers in the overparameterized regime (as authors point out, e.g., lines 246-248), and it could enable a series of future studies. The paper could potentially provide foundational insights with a potentially wide range of applications in understanding the mathematical properties of attention mechanisms in deep learning (though, whether this is achieved or not is not fully clear, as I discuss under weaknesses).

**Weaknesses:**

**Key weakness:** The main weakness of the paper which is present throughout the paper is that the ultimate goal of the theoretical results as well as their implications tend to seem lost and poorly explained. I believe many among the ML audience of this paper might raise the objection that “this paper belongs to a math journal, not ICLR”. I take the liberty to clarify that that is not my objection, and I believe even a paper that solely focuses on deriving results and developing mathematical tools from a purely theoretical perspective could be valuable and could very well belong to ICLR. That being said, I expect such a paper to clarify the implications of their results for the wider community, or at least for future theoretical research on foundations of deep learning. I regret to see that this paper does not meet this criteria. As I mentioned among the strengths, I understand the results are foundational with potentially important consequences, but they are highly (perhaps overly) mathematicized, without proper explanations that clarify what these mathematicized observations entail for the ML applications, e.g., how transformers learn their representation, or how these result allow us to develop a better understand of the convergence of GD-based optimization on transformers (which I think they could do).



**A list of some of the main weaknesses:**

1. The implications of the theoretical statements are often not discussed (with a small number of exceptions). For instance, the main result of the paper is stated in Theorem 3.13, but no discussion of its implication is provided. In an ML paper, all such theoretical results should follow with a discussion of the implications that are more accessible to ML researchers.

2. Even when implications are discussed, the ultimate goal or the consequences of the implications are not discussed. e.g., the authors discuss the implications of corollary 3.5, but they can clarify what this implies for the success of the attention mechanisms, or at least remark how it could enable future studies on that.

3. The experiments lead to a conjecture, but they are from empirical relevance. I would like to clarify again that this is fine with me, as long as at least the theoretical relevance for future work that could eventually lead to an empirically relevant application is discussed.

4. There are too many technical details leading to the proofs or sketch of the proofs of the results, which I do not think belong to the main text of a machine learning paper (e.g., lines 165-173 or 227-240). These details could be move to the appendix and replaced with a more high-level discussion of the intuition behind the results and the implications of the results.



**More specific points about the weaknesses:**

a. There is almost no review of the literature. There are many interesting (and some quite relevant) recent works on developing a theoretical understanding of transformers or relevant topics (e.g. learning dynamics in overparameterized regimes). It is important to discuss this and explain the gap in the literature. This helps motivate the paper as well.

b. While the introduction motivates the paper, the abstract does not. The abstract could be improved to reflect the motivation explained in the second paragraph of the introduction. Moreover, the second paragraph of the introduction itself could mention more specific examples of the applications and the potential future direction the paper opens a door to, in order to better motivate the paper.

c. The purpose of the visualizations in figures 2 and 3 is unclear.

d. Line 297 says “we introduce yet another re-parameterization”. While I understand you later use this, I suggest you explain closer to where you introduce the reparameterization and why you are introducing it.

e. The summary of the proof of Theorem 3.13 could also be moved to the appendix. It is crucial to discuss the implications of the theorem, instead of explaining the proof in the main paper.

f. The presentation can be improved to be more accessible to ML researchers.

**Questions:**

1. The authors say in the conclusion that “Extending our algebra-geometric analysis to include these variations represents an interesting line for future investigation” (line 496). How? Can you be more specific? What do you think are the future direction of the research? Can you specify 2-3 specific directions?

2. What purpose do figures 2 and 3 serve?

3. What do you think are the main implications of the main theorem?

**Details Of Ethics Concerns:**

No ethical concern

---

> ### Author Response · Authors · 2024-11-18
>
> We thank the reviewer for the several comments and suggestions, resulting in constructive feedback. We have re-uploaded an updated version of the manuscript, where we address several weaknesses raised. As suggested, we have moved several technical parts (and a figure) to the appendix, and used the space to motivate and explain the results. In particular, we have elaborated on the importance of dimension and of the fibers in the section Introduction and Related Work, and on the role of singularities in section Single-Layer Identifiability. We have discussed practical implications of our main result (see Corollary 3.8), and extended the section Conclusions and Future Work. Lastly, as suggested in weakness d, we have mentioned that the reparametrization is crucial for stating our main result. Below, we address the questions raised.
>
> Question 1: We agree with the reviewer on the importance of elaborating on possible future extensions. While in our paper we considered a `vanilla’ self-attention mechanism, we believe that it might be possible to extend the theory to incorporate the variations that are used in practice. Two such variations, which are ubiquitous in contemporary Transformers, are **skip connections** and **multiple attention heads**. With both these additions, the lightning self-attention mechanism is still polynomial, which enables to potentially apply similar techniques as in our paper to their analysis. Note that skip connections make the model non-homogeneous, which breaks the scaling symmetry in the parameterization. We conjecture that in this case the parameterization via $(\textbf{M}, L)$ is generically one-to-one, similarly to the traditional case (Conjecture 3.16), which is also non-homogeneous. In contrast, including multiple attention heads introduces new symmetries due to permutation of heads, similarly to the permutation symmetries of traditional Multi-Layer Perceptrons [1]. Therefore, these two variations give rise to interesting phenomena in terms of symmetries of the parameterization, and we believe that these directions are worthy of exploration. We have incorporated this discussion in the section Conclusions and Future Work of the re-uploaded manuscript.
>
> Question 2: Our intent in Figures 2 and 3 is to illustrate via a diagram the statement of Lemma 3.12 and the proof of Theorem 3.13, respectively. The statement of the Lemma involves a subtle recursion, which, in our opinion, is difficult to interpret from equations. The diagram in Figure 2 visualizes this recursion by unrolling it as a tree. The main step in the proof of Theorem 3.13 exploits a symmetry of this tree, which is illustrated intuitively in Figure 3. While we believe it is important to convey an intuition for Theorem 3.13, we agree that the space in the main body can be used to better build motivation and convey significance;  we have moved Figure 3 to the appendix, as suggested.
>
> Question 3: We agree with the reviewer that it is crucial to elaborate on the implications of our main theorem – i.e., the computation of the fibers and, as a consequence, of the dimension of the neuromanifold.  To this end, we have re-uploaded a new version of the manuscript with a discussion on the importance of dimension, especially its relation to **sample complexity** in machine learning, which has  practical implications – see the section Introduction and Related Work.  This discussion is also reported in the collective comment we posted above. Moreover, after Corollary 3.8, we have discussed the implications of our formula for the dimension and, in particular, how the latter compares to the number of parameters, which is commonly used as a measure of expressivity and/or sample complexity.
>
> [1] Kileel et al., On the expressive power of deep polynomial neural networks, 2019.

---

> > ### Comment · Reviewer_qe2M · 2024-11-24
> >
> > I appreciate the authors' thorough response and addressing my concerns.
> >
> > I acknowledge that the implications of the theory are clearer, and given that, I think it is only fair to improve the overall rating. That being said, **I think the presentation still needs improvement**, especially for a paper that receives an overall "accept" rating. Specifically:
> >
> > - I think the clarifications and the discussions of the motivation and improvement in the introduction are helpful, but not sufficiently cohesive.
> >
> > - Some statements in the introduction sound overly elaborate on related topics, and the direct relevance to this work seem unclear (e.g., the last paragraph of the introduction), and general without specified implication).
> >
> > Additionally, since the authors clarify that the implications are particularly relevant for training and sample complexity, I think a discussion (or at least a remark) of optimization in over-parameterized regimes is necessary. This is a topic with a rich body of previous research and of crucial importance for the success of transformers (e.g., previous works on benign overfitting, double descent, implicit bias of gradient descent in over-parameterized model, convergence rate of gradient descent in over-parameterized regimes, etc.). This is particularly relevant given the claims of the authors on the implications of their results for training transformers with the appropriate amount of data (e.g., line 066), which might be misunderstood by an unfamiliar reader; over-parameterization is shown to be critical for the success transformers achieve.
> >
> > On a related note, going back to the weakness **a** I mentioned in my initial review, I think a more through discussion on the relevant literature is necessary.

---

> > > ### Comment · Reviewer_qe2M · 2024-11-24
> > >
> > > To conclude, I raised my overall rating following the improvement in the draft, only to be fair and consistent and reflect the issues the authors addressed. I must express that my assessment is extremely marginal. I hope the authors will make improvements for a final version. Although there is not enough time for additional iterations over the draft, **I strongly urge the authors to improve the presentation and discuss the topics I mentioned above** in future submissions or in the camera-ready version should this paper be accepted.

---

### Official Review · Reviewer_sSei · 2024-11-03

**Soundness:** 4
**Presentation:** 3
**Contribution:** 3
**Rating:** 6
**Confidence:** 4

**Summary:**

The paper studies the geometry of lighting attention models by characterizing their fibers (redundant parameterizations of a neural network) and neuromanifolds (the function spaces induced by all possible parameterizations). Critically, they assume for most of the paper that (1) the attention matrix has no normalization (i.e. no softmax); (2) there are no MLP layers of the transformer; and (3) there are no residual connections. They prove the following:
* From Theorem 3.2, in most cases, the fibers are 1-dimensional (identifiable functions up to rescaling), except under certain rank-1 cases, where the fiber has higher-dimensional symmetries. The dimensionality of these fibers can be used to exactly capture the neuromanifold dimension in Corollary 3.3.
* Theorem 3.4 shows that the neuromanifold is closed in Euclidean space and that its boundaries are defined by networks with rank-1 matrices.
* Theorem 3.15 shows that self-attention layers have singleton fibers, and Conjecture 3.16 predicts that deeper self-attention models also have singleton fibers. The conjecture is backed up with experiments that estimate the dimensionality of the neuromanifold dimension and shows a tight alignment to the conjecture.

**Strengths:**

The contributions reveal interesting properties of the geometry of lightning attention models and provide a rigorous characterization of the resulting functional space. The results are well-described and relatively straight-forward to follow.

As far as I can tell (although I am no expert in algebraic geometry), the claims are clearly stated and the mathematics are correct.

**Weaknesses:**

There are several key gaps between the models studied in this paper and practical models:
* The lack of non-linearities in the attention unit (in particular, the lack of softmax).
* The lack of residual connections between layers.
* The lack of element-wise multi-layer perceptrons between layers.
As far as I am aware, all three components are important for practical transformers, and that a lack of residual connections or MLPs tend to lead to degeneracy in training. While the manifold analysis in the paper is interesting, it's unclear how it would extend to cases where the model does not have such clear polynomial behavior.

While the theory is interesting in its own right, it's somewhat unclear to me what can be inferred about realistic models based on this theory, especially because the softmax results point to no parameter redundancy.

**Questions:**

While the results are certainly mathematically interesting, it's unclear to me what practical guidance can be gleaned from this line of work. What are the consequences of understanding the dimensionality of the neuromanifold for the model's expressivity and generalization?

I think parameter redundancy is an interesting angle, but this work seems to suggest that existing models have little room for improvement on that front. Are there modifications to this work (or questions to be pursued by future work) that might look at notions of fibers that pertain to parameters that map to _approximately_ the same output, rather than exactly the same output?

I appreciate the section that extends this analysis to self-attention layers. Can this analysis be modified to account for the inclusion of residual connections or MLPs?

---

> ### Author Response · Authors · 2024-11-18
>
> We thank the reviewer for the comments and the suggestions. Below, we address the questions raised.
>
> We acknowledge that it is important to elaborate on the importance of understanding the dimension of the neuromanifold. To this end, we have re-uploaded a new version of the manuscript with a discussion on the importance of dimension, especially its relation to **sample complexity** in machine learning, which has  practical implications – see the section Introduction and Related Work.  This discussion is also reported in the collective comment we posted above.
>
> Regarding the question raised on "approximate fibers", we believe that this is an interesting point. As mentioned by the reviewer, this might provide a relaxed identifiability condition, that can potentially inspire improvements in terms of architecture design. One way to formalize this is by considering pre-images $\varphi^{-1}(U)$ via the parameterization map of neighbourhoods $U \subset \mathcal{M}$ in the neuromanifold $\mathcal{M}$ – for example, balls $U = B_{\varepsilon} \cap \mathcal{M}$, $\varepsilon \in \mathbb{R}_{>0}$. Such pre-images contain parameters that are mapped to functions that are close in the neuromanifold, according to some prescribed metric. One issue is that, different from fibers, pre-images of balls do not have a regular behavior: it is not true, for example, that they have the same dimension, for a generic ball in the neuromanifold. The diameter of these preimages is closely-related to – more precisely, it is upper bounded by – the **Lipschitz constant** of the parameterization map. Studying this Lipschitz constant is interesting by itself, since it is related to the regularity of the parameterization, and, as a consequence, to the "flatness" of the loss landscape for a given dataset/task. The Lipschitz constant of a polynomial map depends on the degree of the polynomial, which for deep lightning self attention networks equals to $3^l$, where $l$ is the number of layers (see lines 151-152 in the paper). This suggests that the degree – which is a fundamental invariant in algebraic geometry – might play an important role when considering `approximate fibers’, and might be connected to more general questions on optimization. Therefore, we believe that this is an interesting direction to explore.
>
> We wish to comment on the extension of the result to include skip connections and/or MLPs. First, including skip connections into lightning self-attention layers still results in a polynomial model, but makes it **non-homogeneous**. This breaks the scaling symmetry in the parameterization. We conjecture that in this case the parameterization via $(\textbf{M}, L)$ is generically one-to-one, similarly to the traditional case (Conjecture 3.16), which is also non-homogeneous. We believe that this might be possible to prove with techniques similar to ours, and we have some ideas for approaching this. However, we do not have a complete argument at this stage, and we plan to work on this as a next step. Regarding MLPs, it is possible to include them in the model and potentially apply algebraic techniques similar to ours, as long as their activation function is polynomial – see [1] for an overview of polynomial MLPs. However, the fibers of the parameterization and the dimension of the neuromanifold are not well-understood for polynomial MLPs, and characterizing them is a major **open problem** [2], although recent progress has been made [3]. Even though we agree that this is an important future direction since it reflects the Transformer architectures that are used in practice, we believe that the complete understanding is far away, and this would require significant theoretical advances in order to be addressed. Still, our work could provide a basis for reasonable next steps along this path. In summary, both these directions constitute important lines for future work, and we have incorporated the content of this discussion in the new version of the manuscript (see Conclusions and Future Work), which we have re-uploaded.
>
> [1] Kileel et al., On the expressive power of deep polynomial neural networks, 2019.
>
> [2] Kubjas et al., Geometry of polynomial neural networks, 2024.
>
> [3] Finkel et al., Activation thresholds and expressiveness of polynomial neural networks, 2024.

---

> > ### Comment · Reviewer_sSei · 2024-11-23
> >
> > I appreciate the authors' thoughtful response and their detailed discussions of the topics I mentioned. I appreciate their inclusion in the edited paper.
> >
> > To the response about non-homogeneity of skip-level connections and MLPs, my mental model of them is that this asymmetry is necessary for harnessing the capabilities of transformers. I would further recommend addressing some of the concerns raised by papers like [1], which posit that deep models without these aspects tend towards degeneracy.
> >
> > My score will remain the same, but I thank the authors for their response nonetheless.
> >
> > [1] https://arxiv.org/abs/2103.03404

---

### Official Review · Reviewer_fk8M · 2024-11-03

**Soundness:** 3
**Presentation:** 3
**Contribution:** 2
**Rating:** 6
**Confidence:** 2

**Summary:**

This paper provides a theoretical analysis of the geometry of the function spaces corresponding to self-attention networks. The core of the work concentrates on the “lightning” variant of self-attention (for which forward computation computation is linear in the sequence length) since it is amenable to an algebraic analysis. The authors calculate the dimension of the hypothesis space (“neuromanifold”) through an analysis of the (generic) fibers of the model’s parametrization. The submission provides an extension to normalized attention modules, provably for the case of a single layer, and conjecturally for deep networks.

**Strengths:**

S1. The paper exposition is quite clear, striking a balance between the lingo of algebraic geometry and the context of the ICLR audience.

S2. The empirical popularity of attention-based networks, and the relative lack of understanding on the properties of said architectures (with respect to fully connected and convolutional ones, as the authors point out in L51-53) sufficiently justifies the theoretical exploration presented in this submission.

S3. The authors identify lightning self-attention as a fruitful compromise that is both (1) amenable to their algebraic-geometric analysis and (2) a reasonably realistic neural network module. I believe the presented implicit hierarchy of modules (single-layer lightning self-attention, deep lightning self-attention, single-layer normalized self-attention, deep normalized self-attention) can be of use in further theoretical analyses of attention networks.

S4. The authors introduce a “virtual weights” parametrization which enables an elegant analysis of deep lightning self-attention networks.

S5. The submission poses a conjecture on the fibers for deep normalized self-attention networks (under their virtual weights parametrization). This conjecture is theoretically validated in the case of a single layer and empirically tested for deep networks. The communication of this conjecture to the relevant communities within the context of the ICLR conference can be beneficial towards advancing on its resolution.

**Weaknesses:**

While S2 can warrant sufficient interest from the ICLR community, I think the submission falls short of adequately examining the implications of its findings. I think the authors could have expanded more on the “so what?” question after reaching their (interesting!) core theoretical goals.

W1. For example, in L416-419, the authors are quite descriptive about the dimension of the neuromanifold for a contemporary language model being ~8.2B vs the crude parameter dimension of ~8.6B. Yet, there is not even a high-level attempt to prescribe how to exploit this information.

W2. The authors partially justify the importance of the analysis of the dimension of the neuromanifold via an appeal to learning theory (L58-61) and its connection with sample complexity. (Although this may be considered outside of the scope!) The submission does not present a tangible demonstration of the implications of the knowledge of the neuromanifold’s dimension on the sample complexity, not even in a very simple toy-ish experimental setting.

**Questions:**

Q1 & Q2. What is the authors’ position on W1 and W2?

Q3. In the (much appreciated!) anonymized code for the verification of Conjecture 3.16, L19 instantiates `n_iter_array = [5, 5, 5, 5, 5, 5, 5, 8]`. These iterations are used as “trials” out of which the maximum rank Jacobian is extracted in L41 `dim_vec.append(max(ranks))`. Could the authors explain why such multiple “trials” are needed? And also why is this component left out of the experimental description in L472-484?

Q4. Could the authors elaborate on the implications of the non-smoothness of the $\mathcal{M}_{d, d’, a}$ manifold from an optimization standpoint?

---

> ### Author Response · Authors · 2024-11-18
>
> We thank the reviewer for the feedback and the appreciation. Below, we address the questions raised.
>
> Q1 and Q2: We acknowledge that it is important to expand on the motivation and the significance of computing the **dimension** of neuromanifolds and, in particular, its relation to **sample complexity**, which is a primary motivation behind our main result. To this end, we have re-uploaded a new version of the manuscript with a discussion on the relation between dimension and sample complexity – see the section Introduction and Related Work.  This discussion is also reported in the collective comment we posted above. Moreover, after Corollary 3.8, we have discussed the implications of our formula for the dimension and, in particular, how the latter compares to the number of parameters, which is commonly used as a measure of expressivity and/or sample complexity.
>
>
> Q3: As mentioned by the reviewer, we perform multiple trials when estimating the Jacobian, and then take the maximum. We acknowledge that this might sound superfluous, since taking the rank of the Jacobian at an arbitrary parameter yields the correct result almost everywhere w.r.t. such parameter, as explained in lines 436-438 in our paper. However, in practice this does not hold due to the discrete nature of computations – i.e., the floating-point approximation of real numbers – and due to **numerical errors** when calculating the rank, leading to an underestimate of the desired generic rank. In order to amend for this, we perform a few estimates at random parameters, and take the maximum over the ranks obtained. The resulting estimation is more robust to numerical errors. We did not mention this in the text since we believe that it constitutes a small implementation detail, that also depends on the programming framework – more specifically, on its numerical precision – and that is often unnecessary.
>
> Q4: We wish to additionally comment on the relevance of singularities of neuromanifolds and, in particular, on their relation to the training dynamics. Singularities of neuromanifolds are a central focus in Information Geometry, and specifically in **Singular Learning Theory** [1]. According to the latter, singularities of neuromanifolds play a central role in deep learning, since the function learned by a neural network (via gradient descent) often corresponds to a singular point of the neuromanifolds [2].  In other words, singularities often attract the learning dynamics, resulting in a form of `implicit bias’ associated to the neural architecture. According to our result for a single-layer lightning attention network (Corollary 3.8), singularities of the neuromanifold arise exactly when both the attention matrix $A$ or the value matrix $V$ have rank $\leq 1$ (or vanish). Therefore, this result suggests an implicit bias in attention mechanism towards inferring (extremely) **low-rank** functions. Such bias has been empirically observed in a variety of neural architectures [3], and our result might suggest a mathematical explanation to this phenomenon, at least in the single-layer and lightning case. Since we believe that this discussion is important to include, we have re-uploaded a new version of the manuscript that includes it in the section Single-Layer Geometry.
>
>
> [1] Watanabe, Algebraic geometry and statistical learning theory, 2009.
>
> [2] Amari et al., Singularities affect dynamics of learning in neuromanifolds, 2006.
>
> [3] Huh et al., The low-rank simplicity bias in deep networks, 2021.

---

> > ### Comment · Reviewer_fk8M · 2024-11-19
> > **Response to rebuttal**
> >
> > I thank the authors for their responses to my questions. I appreciate the additions to the manuscript on the motivation and significance of their main results. I maintain my original rating for this submission.

---

### Author Response · Authors · 2024-11-18
**General Reply to all the Reviewers**

We wish to thank all the reviewers for their comments. Multiple reviews share a common concern about the **motivations** and  **implications** of main results. We wish to elaborate on this. To this end, we have re-uploaded a new version of the manuscript containing extended discussions and motivations. The additions are highlighted in **blue**, and we summarize them below.

Our main result is concerned with computing the **dimension** of a neuromanifold. The latter is a precise measure of expressivity, and is closely related to **sample complexity**. According to the Fundamental Theorem of Learning, the dimension controls the sample complexity of learnability [1]. Sample complexity is fundamental in practical applications; an expression for sample complexity can be used both to select the appropriate model/architecture given an available dataset, and to collect appropriate amounts of data to train a given a model. This is especially important for attention-based models, that are nowadays popular in several domains, and are trained at extremely-large scales. We have included a discussion on the relation to sample complexity in lines 59-67 of the re-uploaded version.

We compute the dimension by actually characterizing the (generic)
**fibers** of the parametrization map Theorem 3.17). Intuitively, the theorem states that a generic function in the neuromanifold has precisely three types of symmetries in its parameters: scaling each layer by a constant, scaling the keys and queries of each layer by invertible matrices, and scaling the output of one layer by an invertible matrix and reverting this scaling in the next layer (this more accessible explanation can now be found in our article in lines 258-263 and 354-358). Now, the dimensions of the fibers measure the difference between the dimension of the neuromanifold and the number of parameters. This difference can be large due to the symmetries from Theorem 3.17. In some attention-based models, the number of parameters can be 50 % larger than the dimension – as we elaborate in lines 369-376 of the re-uploaded paper – which is in contrast with other models such as MLPs since their fibers are typically small [5].  Therefore, an expression for the dimension of the neuromanifold gives a precise estimate for the sample complexity, which is instead commonly achieved by counting parameters.

Moreover, understanding the fibers can be interesting beyond their relation to the dimension. Fibers of the parametrization induce invariances of the loss function which are data-independent, meaning that for any dataset, the loss will be constant for parameters within the same fiber. In particular, they give rise to the phenomenon of flatness of the loss landscape [2]. The gradient directions must be orthogonal to fibers of the loss, which also induces a constraint on the Hessian of the loss [3]. This has recently been exploited to design optimizers that `teleport’ along fibers [4], improving learning efficiency. We have incorporated this discussion around fibers in lines 71-82 of the re-uploaded version.

From a broader perspective, the ultimate goal of studying neuromanifolds is to understand how changes in a network’s architecture influence the geometry of the neuromanifold – such as its dimension and singularities – and how, in turn, this geometry impacts fundamental aspects of machine learning – such as sample complexity and convergence guarantees. Our work provides a first step in this direction for attention-based networks, which are ubiquitous nowadays.


[1] Shalev-Shwartz and Ben-David, Understanding machine learning: From theory to algorithms, 2014.

[2] Zhao et al., Symmetries, flat minima, and the conserved quantities of gradient flow, 2022.

[3] Kunin et al., Neural mechanics: symmetry and broken conservation laws in deep learning dynamics, 2020.

[4] Zhao et al., Symmetry teleportation for accelerated optimization, 2022.

[5] Kileel et al., On the expressive power of deep polynomial neural networks, 2019.

---

### Meta-Review · Area_Chair_TuS5 · 2024-12-19

**Metareview:**

This paper studies the geometry of single-layer and multi-layer linear-self-attention networks via the neuromanifold. It reveals interesting properties of the geometry of lightning attention models with a rigorous characterization of the resulting functional space. Besides, the paper exposition is clear, striking a balance between the lingo of algebraic geometry and the context of the ICLR audience. I recommend to accept.

**Additional Comments On Reviewer Discussion:**

After the discussion, the authors have addressed most issues raised by the reviewers, e.g., paper writing, parameter efficiency over 50%.

---

### Decision · Program_Chairs · 2025-01-22

Accept (Poster)